# Street Clinics and the Healthcare of Vulnerable Homeless Communities in Brazil: A Qualitative Study

**DOI:** 10.3390/ijerph19052573

**Published:** 2022-02-23

**Authors:** Giulia Romano Bombonatti, Maria Giovana Borges Saidel, Fernanda Mota Rocha, Débora de Souza Santos

**Affiliations:** School of Nursing, State University of Campinas, Campinas 13083-887, Brazil; grbombonatti@gmail.com (G.R.B.); mgsaidel@unicamp.br (M.G.B.S.); f228274@dac.unicamp.br (F.M.R.)

**Keywords:** homeless population, harm reduction, health services accessibility, vulnerable populations, intersectionality, social discrimination, health status, social conditions

## Abstract

(1) Background: homeless people are characterized by serious social vulnerability and difficulty in accessing health services worldwide. In Brazil, this population is supported by the Street Clinic teams who are challenged to establish intersectoral networks to expand access and promote unique and humanized care from the perspective of harm reduction. The study aimed to analyze the practices of professionals working at the Street Clinic in a city in the interior of São Paulo, Brazil, to tackle the vulnerabilities of the homeless population and expand access to the health care network. (2) Methods: a qualitative study was conducted with a social approach in which we interviewed eight workers from the Street Clinic. Data were analyzed using the thematic content analysis tool. (3) Results: three nuclei of meaning were built: stigma and prejudice as the main barriers to accessing services, harm reduction as a humanized care practice and valuing autonomy, and Street Clinic as a gateway to the health system and main interlocutor with other services. (4) Conclusions: the articulation with network services is marked by contradictory relationships, of conflict and trust, signaling the need for greater investment in educational qualifications and working conditions for professionals at all levels of care to expand access to health care.

## 1. Introduction

Worldwide, homeless persons experience social problems resulting from a financial, mental, cognitive, behavioral or physical situation, sleep in temporary shelters or live in inadequate or insecure housing [1,2].

The high costs of rent, electricity and gas bills are contributing factors leading to the homelessness of families and lack of security. A study in a neighborhood in the United States indicates that unhealthy housing conditions are risk factors for infant mortality and that women experiencing homelessness have a greater chance of high-risk pregnancy, violence, use of psychoactive substances, and reduced access to food and health services [3].

A study in England indicated the scarcity of interventions to support, care for and reduce the health risks of people in this condition, in addition to several barriers to accessing primary health care and social care services, such as stigmas, inflexibility, lack of understanding and poor responsibility on the part of professionals as to following up the cases. The experience of being homeless is already traumatizing, and lack of care contributes to new traumas for these individuals [1].

Homeless persons have complex health conditions, receive little or no support from the community and social services, and have difficulty accessing health services or services that meet their needs. There are reports of homeless persons who feel neglected, stigmatized and discriminated against by health professionals, choosing not to seek help from health services. It should be noted that the intersections of class, race and gender permeate the profile of the homeless population, aggravating discrimination [2].

In the Brazilian reality, homeless persons experience similar issues concerning health conditions and difficulty accessing health services. The barriers to accessing health services include difficulties identifying appropriate units in the health care network, negligence and situations of racial, gender and sexual orientation discrimination in the service units, inadequate approaches and reception, technical and emotional unpreparedness of professionals, bureaucratic complications, and emergency services with inadequate geographic location. In addition, homeless persons seek health services when they are in an advanced stage of suffering and are no longer able to cope with or resist the pain on their own [4,5].

This reality is compounded by the violence suffered by this population. The 2019 Brazilian epidemiological report pointed out that 17,386 (2.2%) notified cases of violence were mainly motivated by the victim’s homelessness. Of these, the highest frequency of notification was for females (50.8%), black and brown race/skin color (54.8%), heterosexual sexual orientation (65.2%), and gender identity of transsexual woman (1.7%). It should be noted that the ignored/blank fields for gender identity (33.6%) and sexual orientation (21.1%) were filled in at a high rate. The main forms of violence notified were physical (92.9%), psychological/moral (23.2%), torture (3.8%), sexual (3.9%), and neglect/abandonment (2.7%) [6].

Brazil’s Unified Health System (SUS) was instituted in 1988 by the Federal Constitution and was the result of a social struggle for equality, democracy and emancipation, establishing health as a right for all and a duty of the state [7]. Primary health care constitutes the main entry point for people to access the SUS and organizes their flow within the health network for comprehensive care, that is, person served are referred to more complex services as needed [8].

In the context of health care for the homeless population and in response to the Brazilian reality, in 2012, through the National Primary Health Care Policy the authorities created Street Clinics: health care devices where care is provided by multidisciplinary teams composed of nurses, psychologists, social workers, occupational therapists, physicians, social agents, nursing technicians or assistants, oral health technicians, among other professionals, with actions aimed at humanized and inclusive care. The service uses the harm reduction strategy for users of psychoactive substances and aims at protecting the mental health of these individuals [9], in addition to supporting their actions while respecting the person’s identity and freedom, offering information that encourages them to exercise autonomy and citizenship [10]. To this end, the teams are guided toward intersectoral coordination and effective, comprehensive, equitable and coordinated network health care [9].

Through discussion of the issue of access to services for such a vulnerable population, we adopted as a theoretical framework for analysis a set of contemporary productions on access/accessibility [11,12,13,14,15,16,17] a concept understood as a group’s capacity to seek and obtain health care. Among the numerous definitions in the literature, we chose those that deal with this concept through various dimensions: “availability” relates to the volume and type of existing services, the volume of users and their health needs; “accessibility” refers to the location of the provision and users, distance, movement, and costs; “functional adequacy” relates to the organization of the providers to accept users and their capacity and skills; “financial capacity” refers to supply and costs; and “acceptability” relates to attitudes of users, health workers and the practices of these services [17].

In conjunction with this perspective of access, we used the expanded concept of vulnerability described by Ayres to analyze the complex situation of homeless people in terms of propensity to illness in its multiple dimensions. The concept of vulnerability incorporates risk, the need to value the context, the voices of people affected, and a non-segmented thinking. Ayres classifies vulnerability into individual dimension (involving aspects of the individual’s way of life), social dimension (understands the health-disease process as social processes that are related to material, cultural, political and moral aspects of life in society), and programmatic dimension (mediated by the social resources that people need, which, in the case of the current study, are health care and social care institutions), which are interdependent and must be connected in seeking to solve problems [18].

Assuming that Street Clinics are configured as powerful devices for addressing vulnerabilities, by expanding access to health care for homeless individuals, this study aimed to analyze the practices of health professionals in Street Clinics to cope with the vulnerabilities of the homeless population and expand access to the health care network.

## 2. Materials and Methods

### 2.1. Participants

The research team consisted of four researchers, who shared tasks in determining the objective, study design and data analysis. The coordinating researcher (D.S.S.) and the main researcher (G.R.B.) met weekly during data collection. There were group discussions to code the data, which took place at first weekly and for adjustments every 15 days, approximately.

This study selected eight (*n* = 8) workers from the multidisciplinary team, who were available for interviews during the data collection period, from a total of 19 workers, who worked in the health unit of the Psychosocial Care Network: Street Clinic, a device created in 2012 in Brazil. A nurse, nursing technician, physician, psychologist, social worker, occupational therapist and harm reducers were selected to share their experiences about the perception as to the access to health services by the homeless population and harm reduction practices. All participants agreed to participate and all completed the data collection steps. Here, it is worth noting that, in Brazil, harm reducers are professionals characterized by performing activities that aim to ensure the care for and defense of people at personal and social risk, and bringing teams closer to the values, ways of life and culture of homeless people. They are trained by the health care network and do not need to be trained in any higher education program [19].

The workers were selected through the composition of a sample of convenience and the sample was closed by theoretical saturation. Participants were invited to participate through meetings held in the unit with the aim of acculturation, a fundamental process for the reliability of qualitative research: the first, in which one of the researchers presented the research project in loco and then participation in practical activities of the multidisciplinary team, in addition to participation in team meetings during the period from May to July 2019. The final sample’s demographic data included professionals of different categories, ages and genders, in order to triangulate the sample with the perceptions of different social agents of the team that constitutes the Street Clinic (Table 1). The data were kept safe under the custody of the authors only. The subjects were treated by code names that do not allow their identification. This study received ethical approval from the Research Ethics Committee of the State University of Campinas under opinion number 3,265,854.

### 2.2. Study Design

This was a qualitative study with an approach aligned with social research in health, which assessed the regularity of the phenomenon and analyzes human expressions in relationships, subjects and representations. This methodology encompasses multidisciplinary and multi-professional discussions, expanding the perspective and approaches to understanding the social group. The study was conducted and reported according to the Consolidated Criteria for Reporting Qualitative Research (COREQ). The study design featured in-person interviews with guiding questions, presented in Table 2, in the form of semi-structured interviews. The research objective was to understand individual experiences about the homeless population, looking into the workers’ perceptions about access and about harm reduction practices. The interview schedule and guiding questions were developed based on the researchers’ academic experience. The semi-structured interviews employed a technique based on dialectical logic, whose methodology allowed for the discussion to have a flow of dialogues, thus enabling the construction of the social group’s perception about the studied object [20].

### 2.3. Procedure

Participants were recruited in the field of study, that is, the Street Clinic unit in a large municipality located in the state of São Paulo, Brazil, during the period of acculturation. Participants were invited in the workplace, with the consent of the local manager, and if they agreed to participate in the interview, they signed the Informed Consent form. The interviews were conducted at the time and place chosen by the participant, during their work activities, and audio was recorded for later transcription into a Word document (Microsoft Office). All recordings and transcriptions were performed with the consent of the participants and stored in Microsoft Office files (Word and Excel), with restricted access, ensuring data property protection.

At the beginning of the interview, participants were informed of their right to opt out of the research at any time without being subject to any consequences. Each interview was conducted with workers individually, without disregarding their belonging to the social group, and lasted a minimum of 11 min and maximum of 38 min. After providing initial explanations (briefing) and obtaining consent, the researcher began with open guiding questions through a semi-structured interview. This procedure allowed for the flexibility of questions during the interview.

The workers shared their experiences, “funneled” towards the object of study, and discussed their experiences of the homeless population’s access to health services, in addition to the daily practices of harm reduction. The themes included in the guiding questions enabled the participants to engage in dialogue on the topic addressed, including: vulnerability and characteristics of the population served by the service; daily activities (potentials and problems); work processes; harm reduction; coordination with Primary Health Care, specialized services of the Psychosocial Care Network and social care services. After each interview, the researcher asked about doubts or further contributions and completed the procedure.

### 2.4. Data Analysis and Feedback

The interviews were conducted by the first author of the article (G.R.B.), who was previously prepared by the coordinator of this study (D.S.S.). The instrument of the interview was based on the following axes: activities, potential and problems of the Street Clinic; characteristics and vulnerabilities of the assisted population by the service; relationship established between the Street Clinic and the homeless population; articulation of the service with others in the network; relationship of other services with the homeless population. Once the interviews were transcribed, to ensure consistency of procedures in all interviews, the two authors above coded the interviews separately. The themes that emerged were then discussed and analyzed by pairs of research groups and analyzed together to determine the final themes. The survey also adopted a structure aimed at providing transparency for the analysis process, which is shown in Figure 1 (concept map).

The transcripts generated a corpus of interviews, which were analyzed using the thematic content analysis technique, emphasizing the identification, analysis and interpretation of meaning patterns (“themes”). It was carried out in a hybrid way, aiming to encompass social theoretical frameworks that make sense for the researched population [20,21]. The analysis consisted of six steps described below, in Table 3.

The thematic analysis of social research goes from descriptive to interpretive and the process consists in categorizing the transcripts into broad themes and, through a careful, systematic and continuous review, translating the data into more specific themes [22]. In qualitative research with a social approach, the analysis focuses on the subjective experiences lived by the participants [20]. In this regard, researchers must demonstrate reflexivity throughout the study, meeting a principle of credibility of qualitative research [22]. This reflexive process encourages researchers to consider their worldviews and how this can impact the research process, as the researcher becomes an integral part of this dialectical logic in qualitative research; however, it is necessary to encompass reliability and credibility criteria of qualitative studies. Thus, this type of approach meets the objective of study of this research, particularly with regard to data analysis, since the analysis processes protect from this interference that could affect the results of qualitative research [23].

After analyzing the data, the first author gave feedback to the Street Clinic workers. This space was built during a team meeting and the results were discussed in the presentation and conversation round format. In this process, the professionals validated the data found and they had the opportunity to revisit the perceptions and expand the problematization. Complete final reports were sent to the service. In November 2019, the results were also presented at the institution that financially maintains the Street Clinic unit, through invitation. On that occasion, the research was presented to the institution aiming to “promote and value the exchange of experiences and production of knowledge in mental health”.

## 3. Results

Analysis of the professionals’ statements resulted in three main thematic nuclei being built to present the results: “The complexity of those who live on the streets”, “The work of Street Clinics from the perspective of harm reduction”, and “Beyond the Street Clinic: network relationships”. Each nucleus was divided into sub-items for better understanding, presented below and exemplified with the statements of the interviewed professionals.

### 3.1. The Complexity of Those Who Live on the Streets

Individual vulnerability and the singularities and “repeated” trajectories

The sub-item refers to the individual characteristics of homeless people, their diversities, particularities and trajectories. Despite their singularities, history repeats itself: many left or were expelled from home because of the intense use of psychoactive substances and, being homeless, they continue or start using them; others were expelled due to their sexual orientation and gender identity; and many on account of unemployment.


*“People are there in the square, also because we put them there, because we see that there are several trans, gay, and homosexual girls who are homeless, because their parents threw them out. Then they go live on the streets and lose everything, because what they look for all the time on the streets is this: family, understanding, acceptance, respect, which they don’t have at home, so this is difficult.”*

*(2)*


Social vulnerability and violence

This sub-item is marked by the invisibility of the population to society and by everyday violence. In this context, women stand out, as they are physically and sexually assaulted by partners, resulting in unwanted and risky pregnancies. The following statements exemplify these oppressions:


*“The Cathedral is the largest church in the city, where many [homeless] people stay sleeping nearby. And one way to meet the request of the churchgoers, who enter the Cathedral and say that the church door is always smelling of urine, poop, was to wash the church’s sidewalk. And they were doing that at dawn, in winter. And then blankets got wet, and then they went to sleep wet on the wet blanket, cold, then several of them were with pneumonia, several were hospitalized, several of them died.”*

*(2)*


Programmatic vulnerability and the difficulty of access

The sub-item elucidates the institutional and access barriers of the homeless population. The care provided by health professionals full of stigma and prejudice culminates in late access to health services, with problems being intensified.


*“They [homeless population] are very badly treated when they arrive at the hospital, in many ways they are neglected, abused, left aside, it’s a life that we realize that is not worth investment in both in terms of health and all other other spheres of [social] care.”*

*(2)*



*“You already see the extreme condition, you know, you don’t get it from the beginning, you get a diabetic with diabetes at 600, you can hardly identify it, the device doesn’t read it.”*

*(5)*


### 3.2. The Work of the Street Clinic from the Perspective of Harm Reduction

Harm reduction

The sub-item emphasizes the fact that professionals understand the harm reduction strategy as the central axis of work in Street Clinics. According to the professionals, this work leads users to important reflections about their health and decision-making processes in the situations of use. Harm reduction respects individuality, freedom of practice, the choice to reduce use or stay abstinent, and guarantees access to care.


*“This pipe, you won’t share it, because your mouth has several cracks, so you can get tuberculosis, hepatitis, you can get some other disease, that’s why you can’t share your pipe. You have your lip balm and it’s yours, yours alone, because it treats the chapped lips.”*

*(5)*


Street Clinic activities

The sub-item explains how the work process of the Street Clinic is conducted, being divided by the interviewees into activities in fixed fields, mobile fields and team meetings.


*[Fixed field]: “We have our work process, which we call fixed fields, in which we use our own work apparatus, our tent, we have a van, you know, that takes us to these places we call field, where we are going to provide health care, based on our clinical practices.”*

*(3)*



*[Mobile field]: “We go to places in the municipality to conduct activities that can be harm reduction, or we can even go there with no plan, for whatever occurs, we go around some regions of the territory there to be able to access some users; some of them we have already met, we try to maintain a bond there in order to maintain regular care.”*

*(3)*



*[Team meeting]: “We try to carry out this collective construction of clinical cases, I think that what mostly determines this are our meetings with users in the field, where we are be able to build bonds and build a history of care and treatment with them.”*

*(3)*


Building the bond

The sub-item shows that the relationship between professionals, users and service practices contribute to expand access. Thus, the bond is used as a strategy for fostering the


*“We suffer with them, feel their anguish, because that’s it, the contact of being very close, because we don’t have barriers, walls, we’re here, in the open, so we’ll touch, we’ll talk.”*

*(8)*


### 3.3. Beyond the Street Clinic: Networked Relationships

Partnerships

The sub-item emphasizes that matrices, referrals and counter-referrals are essential for effective and comprehensive health care, provided they are implemented in a network. Professionals seek effective communication so other services can better understand the needs of the homeless population, thus providing different services and sharing information to properly manage the most complex cases and understand the particularities of this population.


*“Our work will only work if we have a network formed, alone is no use either.”*

*(8)*


Conflicts

The sub-item underlines the bureaucratic process of social care services as negative in the context of network coordination. As for health care services, the interviewees revealed several conflicts, as some health units refuse to provide care or do so with signs of negligence.


*“The impression we get is that it’s as if it were our responsibility, for being homeless, only ours, only ours.”*

*(5)*



*“Perhaps because some services do not know us, our work, they do not believe in partnerships, they may believe that we take the user there, and then we will not continue our care on the street.”*

*(7)*



*“Sometimes we are invisible with them [homeless people].”*

*(8)*


Coping

The strategies used to cope with the difficulties that guarantee the homeless population have access to services were individual or collective, but not institutional. Skills such as insistence, resistance, resilience and struggle for visibility stand out.


*“There are great partnerships with some services, others we even go there for confrontation, we take the bull by the horns and try to hold our ground to make them understand that they really have to care for them, that there are people here.”*

*(1)*


Network building

The sub-item highlights the lack of implementation of effective paths in building relationships in a network of partnerships.


*“It doesn’t mean that everyone has to have the same opinion, but they should produce a certain alignment and a certain coherence of longitudinal care, so what we call network health care—and this happens a lot in the field of micropolitics, I say that because, you know—it will depend a lot on who is in what place and in what situation.”*

*(3)*


## 4. Discussion

Aiming to analyze the practices of health professionals in Street Clinics, using interviews for data collection and thematic content analysis, our results showed how daily practice at the Street Clinic characterizes the experience of workers in caring for the Homeless Population. The work processes present a health care logic that often does not consist of the health care provided by other units and to other population profiles. This fact poses one of the main challenges for professionals who work in this strategic device of the Unified Health System. The workers provided an account of how they undergo the complex experience of health care for the homeless population and were able to perceive the particularities of health care for this population.

### 4.1. The Complexity of Those Who Live on the Streets

The barriers faced by the homeless population in accessing their rights put them in a situation of vulnerability. We base the concept of vulnerability on the understanding that these people are more susceptible to suffering harm due to the disadvantage in enjoying citizen rights, therefore being subject to social injustice [24]. For further discussion and organization of the results, we considered that the vulnerabilities should be divided into individual, social and programmatic dimensions [18].

The workers find complex situations in the homeless population that must determine the care provided, among which the following stand out: family situation; sexual identity; gender; and race. These situations intensify vulnerabilities in all of their dimensions, as they intersectively impact the marginalization and stigmatization of this population [25].

Family support is considered a protective factor in the individual dimension of young people who belong to sexual and gender minorities. The absence of this support, or even the rejection of the family and, in some cases, the expulsion of the individual, trigger negative effects on the well-being of this population, especially with regard to mental health [5,26]. Black, sexual minority, and gender minority people and homeless people have a high vulnerability to experience difficulties and emotional suffering; this population also has more problems related to the abuse of psychoactive substances [27] and exposure to violence [28,29]. Studies conducted in Brazil and Chile have shown that these people are made invisible by society, as their rights are denied, including the right to belong. There is a need to label this population in order to further marginalize these bodies that are hidden by the social structure [30,31,32].

The study of prejudice and discrimination in health to ensure equity in access to health services is considered a priority in the world’s governmental agendas. The multiple manifestations of discrimination and prejudice are no longer seen as “natural consequences” and are now understood as social problems that must be faced. This statement has an important meaning for contemporary professionals: the identification of behaviors at both the individual and collective levels that are based on racist and sexist explanations—which segregate, neglect and exclude this population—must be faced [33].

The continuation of treatments is another difficulty that workers face in their daily work routines, which leads to the aggravation of the problem. These particularities highlight the need for interdisciplinary health care, that is, co-responsibility for work processes, in which the commitment to the health of the homeless person must be shared and decisions must be taken in a horizontal manner. Accordingly, Street Clinic teams are strategic to guarantee access for this population [34,35].

### 4.2. The Work of the Street Clinic from the Perspective of Harm Reduction

Other important findings refer to the work processes of Street Clinic teams. In Brazil, the Harm-Reduction Policy, despite being under attack by the federal government, is still maintained through the activity of important movements in mental health. Harm-reduction practices are one of the strategies that are adhered to more commonly by homeless psychoactive substance users and are very powerful in building and maintaining the bond between health units and the population. These practices are adapted to the biopsychosocial needs of the subjects, respecting their autonomy and being adjusted to local dynamics, thus placing the subject as the protagonist of their life, with responsibilities and decisions [31,36].

Harm-reduction practices are diverse and are planned according to the needs of the subjects, who are mostly accessed in loco: of these, we highlight the provision of water for hydration, pipes for individual use of cannabis and crack cocaine, lip balms, adequate paper for smoking and cocaine, male and female condoms, and lubricating gel for sexual intercourse. In addition to the inputs, they provide health education aimed at avoiding morbidities and promoting health.

It is noted that this is a public policy maintained by the Brazilian State and implemented by the Unified Health System, and that in recent years, because of ideological political clashes, it has been scrapped through insufficient funding and a decrease in workers specialized in this practice [37]. Nonetheless, practices contrary to the logic of harm reduction and in disagreement with the humanization and comprehensiveness of health care are observed in different parts of Brazilian territory. A 2012 study conducted in the state of Rio de Janeiro reports violent actions by police officers and agents of the Social Care Department, removal of users of psychoactive substances, and compulsory hospitalization [38]. This type of action, endorsed by the state, causes distrust in homeless people, which directly impacts the bond with workers. Hospitalizations do not promote care or cordiality, only violent, custodial and hygienist actions that do not even remotely solve the needs of these individuals who are considered on the sidelines of society [39].

In relation to teamwork processes, the importance of built physical spaces for meetings, training and case discussions is highlighted as, in the daily routine, workers remain on the streets. These spaces are introduced as an antidote to the fragmentation of work and as possibility for expanding discussions on complex cases, establishing bonds, and conducting decision-making. In some services, the headquarters of these units becomes a reference for this population, guaranteeing the right to access. These teams are able to identify aspects of the logic of health care production that are invisible to other units and health professionals, such as the relational dynamics of individuals in the territory and singularities for the organization and adaptation of the work process [40]. Other activities, pointed out by the workers and supported by the literature, are the workshops carried out in loco, called “fixed field,” which use art as a means to deal with anxieties and build therapeutic relationships with quality listening. These activities—in addition to providing important meetings between homeless persons and workers—foster the strengthening of bonds, which is essential for humanized care [31].

All these moments that are part of the work of Street Clinics should aim at the construction of the Singular Therapeutic Project; this is an instrument built in a team that considers the important particularities for a multidisciplinary health care in compliance with the SUS principles and ensures guidelines for the activities and consequently for the work processes. This construction should be shared whenever possible, because, when teamwork promotes integration between the professionals and the population, the results of interventions are enhanced and present effective resolutions [31,41].

### 4.3. Beyond the Street Clinic: Networked Relationships

Finally, by discussing the last category, we can understand that Street Clinic health units provide innovations toward a new health care logic for a population that needs to be cared for in a unique way. In order to understand and manage to meet the complex needs of this population, it is essential to have coordination with the health network as a whole, covering other levels of complexity that are often necessary to deal with the health conditions of those who are homeless [40].

Throughout the study, this network is evidenced as indispensable so such people fully access their rights. The coordination with the other units can be called a network relationship, requiring that it be a referral and counter-referral relationship, that is, the Street Clinic being able to refer some cases and other units providing information about the case for follow-up care [42]. In the everyday routine, unfortunately, this is a practice that is still under construction, which has been emphasized by social agents involved in the care of these users.

Network relationships are configured with many partnerships, but also many conflicts, which characterizes what we call functional adequacy [17]; thus, the health services that are part of the health care network understand or not the users’ needs and, in this process, they can adapt or hinder the provision of care and access. These network relationships require confrontation in the political sphere and with managers involved, as in many situations the necessary coordination is lacking and the service is left with isolated processes.

This reality shows the challenges of overcoming the “barriers” to communication in order to establish an aligned and coherent flow to consolidate the health care network, making the work shared and more effective. The fact that Brazilian Primary Health Care is organized according to fixed residences, by defining areas of coverage, poses one of the greatest institutional challenges for the care of homeless people. This configuration results in a vulnerability in health care practices, and thus hides these individuals from the system in many situations [40].

In the interviews, the workers mention the need for “struggles” both at the individual and institutional levels. These conflicting relations corroborate with a study carried out in the in the state of São Paulo, which found that the “struggle” is directly related to deficiencies in partnerships in personal and intersectoral relationships, causing greater lack of care and often breaking the bond [35].

The main study limitation was related to the fact that the Street Clinic “Consultório na Rua” is a relatively new institution in the health care network; in addition, currently for political reasons, Brazil is facing underfunding in several services. The Unified Health System has restricted human resources available and reduced work teams.

The setting for data collection can also be pointed out as a limitation, due to this being a place with diverse and very dynamic work processes. While the uncontrolled setting is important for qualitative studies, these contingencies may have limited the potential for data collection in two central points: the need of adaptations during data collection, that prevented the researchers from having the time to conduct more in-depth interviews; and the lack of availability of workers that would allow the sample size to be expanded.

On the other hand, the results of the present study have the potential to support discussions and encourage more research on work processes with vulnerable populations. These studies would help to include the need for training human resources with a specific care logic for this population. In our country, unfortunately, research on the care of vulnerable populations is still scarce and has limited funding for its execution. Therefore, this highlights the originality of our study, and how powerful it is to generate new research questions that can contribute both to public policies, training of human resources and the logic of Continuing Education.

Furthermore, the results reinforce the potential of this device; Street Clinics have many characteristics that make it necessary for the logic of health care geared toward homeless persons. The processes of acculturation of workers, the construction and maintenance of bonds with users, the dedication to cases that are mostly complex and challenging, and the struggle to guarantee the autonomy of these people are some characteristics that emerged in this study.

Street Clinics are transformed by experiences and practices and are transforming elements in the health–disease process [41]. The workers recognize the homeless population as individuals, complex human beings who need comprehensive and equitable health care. Among barriers, “struggles” and resistance, these units constitute the main entry point for this population to access the health care system and, therefore, becomes a fundamental mediator of the Psychosocial Care Network [31,34]. Thus, there is evident need for investing and for developing public policies that mobilize managers in favor of this health care logic and not governments and institutions that reinforce the stigmatization of this population.

We also emphasize the responsibility of the other international governments in supporting and promoting practices aimed at caring for the homeless population in their territories, considering their specificities and vulnerabilities.

## 5. Conclusions

The study carried out in Brazil enabled us to identify Street Clinics as devices of the Brazilian public health system that operate as entry points to primary health care for homeless people living on the sidelines of society and suffering high social vulnerability.

The service faces difficulties operating due to its invisibility to other services. The abuse and dependence of psychoactive substances by homeless people, in addition to government policy changes related to drug management as a matter of police coercion and not a public health problem, are challenges on the agenda, since freedom and individual autonomy are jeopardized in a process of public policy dismantling by the current government.

Thus, it is necessary to develop new strategies to address these issues. Teaching about the particularities of the homeless population and harm reduction is essential in continuing education and in higher and technical education in the health area. Paths for the realization of this relationship must be built and planned at all levels of the state, aiming to guide professionals to effectively expand the access to health care for people in extreme vulnerability.

Finally, we hope that the experience of this device developed for the health care of homeless people in Brazil may inspire other countries to build specialized health care services for the needs of a population that is vulnerable and faces all kinds of barriers to access equity healthcare.

## Figures and Tables

**Figure 1 ijerph-19-02573-f001:**
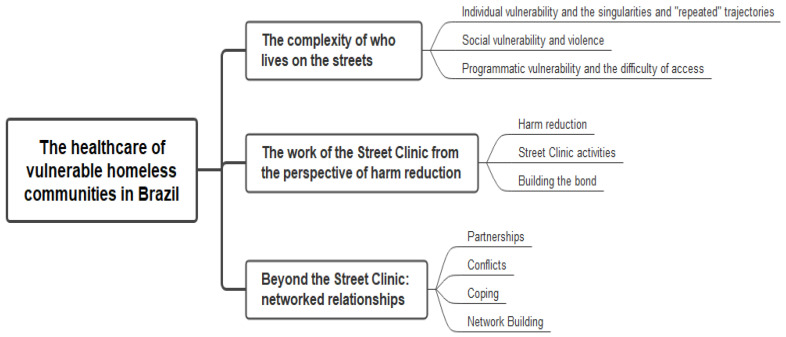
Concept map to view the analysis of results. Brazil, 2021.

**Table 1 ijerph-19-02573-t001:** Participant characteristics. Brazil, 2021.

Id	Gender	Age	Profession	Time of Experience in CnaR (Years)
1	Male	38	Harm Reducer	7
2	Female	45	Nurse	7
3	Male	41	Psychologist	4
4	Male	36	Physician	4
5	Female	35	Nursing Technician	4
6	Female	53	Social Worker	7
7	Female	43	Occupational Therapist	6
8	Female	43	Harm Reducer	7

**Table 2 ijerph-19-02573-t002:** The semi-structured interview instrument. Brazil, 2021.

	Script
1	How does it work and what are the daily activities of the Street Clinic?
2	How is your relationship with the homeless population?
3	What are the main characteristics and vulnerabilities of the population?
4	In your opinion, does the Street Clinic meet the needs of the homeless population in its place of work? Comment.
5	How does harm reduction occur in the Street Clinic?
6	How is the Street Clinic articulation with the healthcare, mental health and social assistance network?
7	What are the strengths and problems of the Street Clinic?
8	In your opinion, what is need to maintain or improve health care for the homeless population, considering the articulation with other services, in addition to the Street Clinic, and comprehensive and humanized care?
9	How is the assistance provide by other services in the health network in relation to the homeless population when the Street Clinic contacts or forwards them?
10	Do other services know how to deal with the issue of alcohol and other drugs? Do you know how to deal with abstinence from patients? Do you apply harm reduction?

**Table 3 ijerph-19-02573-t003:** Six steps of the analysis. Brazil, 2021.

	Steps		Description
1	Pre-analysis	Test skimming and rereading	The research team becomes familiar with the interviews, the concepts emerging in the transcribed corpus
2	Exploration	Coding	Identification of codes that, organized, constitute the initial themes.
3	Clustering	Organization and systematization of codes	Themes and sub-themes are grouped into common categories through both implicit and explicit relevance.
4	Exploration	Interaction	During the interactive process, which involves several reviews, processes for peer verification of themes and sub-themes are included.
5	Treatment	Account	The theme that emerged from the research corpus develops into a narrative (context units) based on the findings. Accordingly, there is the beginning of the triangulation of accounts and the use of citations as units of record (discourse fragments) to illustrate the themes that emerged from the analysis.
6	Treatment	Contextualization	Researchers interpret findings within the context of the existing literature.

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
