# Peer review of "Street Clinics and the Healthcare of Vulnerable Homeless Communities in Brazil: A Qualitative Study"

_ijerph, 2022, doi:10.3390/ijerph19052573_

Round 1

Reviewer 1 Report

I thank the authors for the opportunity to review this interesting article, it is an interesting article; It is a solid study in the methodological approach; but with important inconsistencies in the data collection related to the sample size and the duration of the interviews. The authors comment that there were interviews that lasted between 38 - 11 minutes, the duration seems very short, especially those that lasted 11 minutes. I have my doubts that at that time they deepened the perception of the professionals or that the recruitment of the participants was inadequate according to their ability to contribute. Similarly, having defined the selection criteria for the sample would have been useful in collecting the data. According to what criteria were some participants selected and others rejected?

It would be advisable to delve into the sample size, given that the number of professionals was only 19 workers, why did they not consider including all of them? On the other hand, even though it is a qualitative study, its sample size is small. It would have been valuable if they complemented the data collection with other sources such as personal notes and field notes. It would have been interesting to know the script of questions that they used in the interviews, to know the areas of research raised and the questions that were asked.

In collecting data from the interviews, they comment that they used a technique based on dialectical logic to build the perception of the social group. How were they able to do it with the interviews? Interviews allow limited interaction between the researcher and the participant, unlike what happens in discussion groups.

On the other hand, I believe that the data within the study has not been handled with sufficient confidentiality. They comment that the feedback of the data was shared in the group, why did they do it this way and instead the data collection was done through individual interviews? Don't you think they could reveal personal aspects among the participants? Don't you think that doing it this way violates the privacy of the participants? They comment that they shared the results with the institution that economically hires the professionals. Could it not influence the employment relationship of the workers? Were the workers previously aware that the results of the study would be presented at the institution where they work?

In the analysis section, comment that a thematic analysis will be carried out; while in the results it will be a discourse analysis, it is not very clear to me what kind of analysis they carried out.

Finally, it would be interesting if they had indicated the limitations of the study and future lines of research based on the data obtained.

Thanks a lot. All the best 

Author Response

Firstly, many thanks for your value notes about the paper, that certainly will contribute to the improvement of the text. 

As the English revision was recommended, we submitted the manuscript for another translation with a reputable academic translation company, recommended by our graduate program at Universidade Estadual de Campinas (UNICAMP).

Below are the answers that explain the point by point of the notes.

“It is a solid study in the methodological approach; but with important inconsistencies in the data collection related to the sample size and the duration of the interviews. The authors comment that there were interviews that lasted between 38 - 11 minutes, the duration seems very short, especially those that lasted 11 minutes. I have my doubts that at that time they deepened the perception of the professionals or that the recruitment of the participants was inadequate according to their ability to contribute. Similarly, having defined the selection criteria for the sample would have been useful in collecting the data. According to what criteria were some participants selected and others rejected?”

Reply: In order to sort out the inconsistencies mentioned, some adjustments were made to the text, which are highlighted, as directed. Regarding the time of the interviews, the data collection setting needed adaptations and flexibility, as the interviews were carried out during the work processes. Despite recognizing the importance of the moment of data collection being reserved and uninterrupted, what was possible was carried out within the field, preserving rigor and maintaining the necessary seriousness.

About the sample, the number of participants in qualitative research is composed by the need to maximize the understanding of the phenomenon and the quantity is not a criterion for this type of methodological design. The service consisted of 19 workers, but only 8 workers were available to participate in the interview. Therefore, the final data collection sample consisted of 8 workers.

“It would be advisable to delve into the sample size, given that the number of professionals was only 19 workers, why did they not consider including all of them? On the other hand, even though it is a qualitative study, its sample size is small. It would have been valuable if they complemented the data collection with other sources such as personal notes and field notes. It would have been interesting to know the script of questions that they used in the interviews, to know the areas of research raised and the questions that were asked.”

Reply: The sample is small, as the service in question is small. The semi-structured interview instrument contained the following script, not necessarily being carried out in the same order or words, but they were inserted in a timely manner during the interview:

(1) How does it work and what are the daily activities of the Street Clinic?

(2) How is your relationship with the Homeless Population?

(3) What are the main characteristics and vulnerabilities of the population served?

(4) In your opinion, does the Consultório na Rua meet the needs of the Homeless Population in its place of work? comment

(5) How does harm reduction occur in the street office?

(6) How is the Consultório na Rua articulated with the healthcare, mental health and social assistance network?

(7) What are the strengths and problems of the Office?

(8) In your opinion, what is needed to maintain or improve health care for the Homeless Population, considering the articulation with other services, in addition to the Street Clinic, and comprehensive and humanized care?

(9) How is the assistance provided by other services in the network in relation to the Homeless Population when the Street Clinic contacts or forwards them?

(10) Do other services know how to deal with the issue of alcohol and other drugs? Do you know how to deal with abstinence from patients? Do you use Harm Reduction?

A summary of the main points approached in the semi-structured interview instrument is presented in the last paragraph of the method (line 171 - 179).

“In collecting data from the interviews, they comment that they used a technique based on dialectical logic to build the perception of the social group. How were they able to do it with the interviews? Interviews allow limited interaction between the researcher and the participant, unlike what happens in discussion groups.”

Reply: As described in lines (124-134), articulations were made with the service and workers before the interviews were carried out. The researchers approached the group through interaction and monitoring of the daily service in order to better understand the dynamics and improve the quality of interviews. These procedures disseminated in social research with a qualitative design contribute to the entry of researchers into the field. 

“On the other hand, I believe that the data within the study has not been handled with sufficient confidentiality. They comment that the feedback of the data was shared in the group, why did they do it this way and instead the data collection was done through individual interviews? Don't you think they could reveal personal aspects among the participants? Don't you think that doing it this way violates the privacy of the participants? They comment that they shared the results with the institution that economically hires the professionals. Could it not influence the employment relationship of the workers? Were the workers previously aware that the results of the study would be presented at the institution where they work?”

Reply: The study design was shared with the group before the start of data collection, this occurs because the logic of this health equipment is a logic of articulation and sharing. As for the feedback, in addition to being a request from the group of workers, and an important point for validating the qualitative data, they were carried out by thematic, thus preserving individual perceptions and, consequently, the location and privacy of the participant itself. All care was taken so that feedback to both workers and service managers were carried out in line with the logic of service improvement, which is public and guarantees the protection and job stability of the professionals in the institution.

“In the analysis section, comment that a thematic analysis will be carried out; while in the results it will be a discourse analysis, it is not very clear to me what kind of analysis they carried out.”

Reply: For data analysis thematic content analysis was used. Reference to discourse analysis was not evidenced in the text. A step by step of the thematic analysis was described in Table 2 (line 198).

“Finally, it would be interesting if they had indicated the limitations of the study and future lines of research based on the data obtained.”

Reply: We completely agree with the suggestions of the reviewer in these points. The excerpts about limitations of the study and future lines of research requested have been inserted into the manuscript in lines 525 to 541.

Thank you!

Reviewer 2 Report

Dear authors, thank you for this important and interesting study. Results and discussion still need focus and should be shortened to be more readable. 

Lines are not numbered!

Introduction

3rd Paragraph “it is worth mentioning…” : what about the /?

5th paragraph: “doctor” a medical doctor? GP? Familiy doctor? Other Specialties?

Methods

How were the 8 people selected? Randomized by profession? Did they have volunteered to take part?

Who can you guarantie, that you have a saturation of content by only asking 8 persons?

Who designed the interview-questions? Has it been piloted? How many persons did the interviews? Was it one of the authors?

Page 5 line 1: there is one sentence twofold.

Please describe the research-team: how many persons? Did they coded together or alone? How often did they meet?

Page 5, bullet points 1.6: please move them to a flow-chart or a box.

Page 5: please move the paragraph: “the interviews were coded by”…to the beginning of the methods-section.

Please describe the data management and the handling of anonymizing.

Results

Although very interesting, the result-section is rather long, I suggest to leave out some of the quotes.

Discussion

Please use other discussions from articles you have cited as a guide for length, structure, and content. Please start the discussion with a summary sentence about the study, its method, the main findings. Please include subheadings in the discussion. What are the limitations of the study? The discussion also should be shortened.

Conclusion

The summary should also be shorter and include some specific points about what areas should be explored further.

Author Response

Firstly, many thanks for your value notes about the paper that certainly will contribute to the improvement of the text. 

Below are the answers that explain the point by point of the notes.

Introduction

“3 Paragraph “it is worth mentioning...” : what about the /?”

Reply: Regarding the “/” sign, we chose to use two analogous terms to improve readers' understanding of the unfilled space in some data on the notification form (line 61).

“5 paragraph: “doctor” a medical doctor? GP? Familiy doctor? Other Specialties?”

Reply: Corrected term in text with new English revision (line 75).

Methods

“How were the 8 people selected? Randomized by profession? Did they have volunteered to take part? Who can you guarantie, that you have a saturation of content by only asking 8 persons?”

Reply: About the sample, the number of participants in qualitative research is composed of the need to maximize the understanding of the phenomenon and the quantity is not a criterion for this type of methodological design. The service consisted of 19 workers, but only 8 workers were available to participate in the interview. All respondents participated voluntarily.

“Who designed the interview-questions? Has it been piloted? How many persons did the interviews? Was it one of the authors?”

Reply: The interview script was developed by the main researcher (G.R.B.) and coordinating researcher (D.S.S.), as described in line 180. The PhD researcher  D.S.S. has previous experience with studies aimed at street clinic "Consultório na Rua" services in other Brazilian contexts and used her experience to guide the construction of the research instrument . No pilot was applied due to the difficulty of recruiting participants for the service organized in the street. The interviews were carried out by the first author, after insertion in the field, supervised by the last author. This description was inserted in the text in line 180.

“Page 5 line 1: there is one sentence twofold.”

“Page 5, bullet points 1.6: please move them to a flow-chart or a box.”

“Page 5: please move the paragraph: “the interviews were coded by”...to the beginning of the methods-section.”

Replay: Fixed in text (lines 180 to 198).

“Please describe the research-team: how many persons? Did they coded together or alone? How often did they meet?”

Reply: The research team consisted of four researchers, who shared the objective, study design and data analysis. The coordinating researcher (D.S.S.) and the main researcher (G.R.B.) met weekly during data collection. There were group discussions to code the data, which took place at first weekly and for adjustments every 15 days, approximately.

“Please describe the data management and the handling of anonymizing.”

Reply: The data was kept safe under the custody of the authors only. The subjects were treated by code names that do not allow their identification.

Results

Although very interesting, the result-section is rather long, I suggest to leave out some of the quotes.

Replay: Fixed in text.

Discussion

Please use other discussions from articles you have cited as a guide for length, structure, and content. Please start the discussion with a summary sentence about the study, its method, the main findings. Please include subheadings in the discussion. What are the limitations of the study? The discussion also should be shortened.

Replay: Fixed in text. We completely agree with the suggestions of the reviewer in these points. The excerpts about limitations of the study and future lines of research requested have been inserted into the manuscript in lines 525 to 541.

Conclusion

The summary should also be shorter and include some specific points about what areas should be explored further.

Replay: Fixed in text. We completely agree with the suggestions of the reviewer in these points. The excerpts about limitations of the study and future lines of research requested have been inserted into the manuscript in lines 525 to 541.

 We hope to have attended and answered the main questions of your important review. 

Many Thanks!

Round 2

Reviewer 1 Report

I thank the authors for the work done to improve it, I would only recommend:
Present the questions asked in the interviews using a table in the text, in order to contextualize the readers.
The following considerations should be mentioned as study limitations:
• The adaptations they made during data collection, preventing them from having the time to conduct more in-depth interviews.
• The lack of availability of workers that would allow expanding the sample size
A cordial greeting

Author Response

Firstly, again many thanks for your value notes about the paper, that certainly will contribute to the improvement of the text. 

Below are the answers that explain the point by point of the notes.

"Present the questions asked in the interviews using a table in the text, in order to contextualize the readers."

Replay: Table inserted in the text (page 4).

"The following considerations should be mentioned as study limitations:

  • The adaptations they made during data collection, preventing them from having the time to conduct more in-depth interviews.
  • The lack of availability of workers that would allow expanding the sample size"

Replay: Considerations were inserted in the text (page 12).

Best Regards.

Reviewer 2 Report

Please do not only answer the questions, but include them into the Methods-Section:

“Please describe the research-team: how many persons? Did they coded together or alone? How often did they meet?”

Reply: The research team consisted of four researchers, who shared the objective, study design and data analysis. The coordinating researcher (D.S.S.) and the main researcher (G.R.B.) met weekly during data collection. There were group discussions to code the data, which took place at first weekly and for adjustments every 15 days, approximately.

“Please describe the data management and the handling of anonymizing.”

Reply: The data was kept safe under the custody of the authors only. The subjects were treated by code names that do not allow their identification.

The discussion part is still too long in my poinion.

Author Response

Reviewer 2

Firstly, again many thanks for your value notes about the paper, that certainly will contribute to the improvement of the text. 

Below are the answers that explain the point by point of the notes.

Point 1: “Please do not only answer the questions, but include them into the Methods-Section:

“Please describe the research-team: how many persons? Did they coded together or alone? How often did they meet?”

Reply: The research team consisted of four researchers, who shared the objective, study design and data analysis. The coordinating researcher (D.S.S.) and the main researcher (G.R.B.) met weekly during data collection. There were group discussions to code the data, which took place at first weekly and for adjustments every 15 days, approximately.

Reply 2: Information above inserted into the text (page 3, section 2 of Methods )

Please describe the data management and the handling of anonymizing.”

Reply: The data was kept safe under the custody of the authors only. The subjects were treated by code names that do not allow their identification.

 Reply 2: Information above inserted into the text (page 3, section 2 of Methods )

Point 2: “The discussion part is still too long in my poinion.”

Reply: We revised again and shortened the discussion without losing the arguments. 

Best Regards.
